# Antimicrobial Resistance Profiles and Genes of Staphylococci Isolated from Mastitic Cow’s Milk in Kenya

**DOI:** 10.3390/antibiotics10070772

**Published:** 2021-06-24

**Authors:** Christine M. Mbindyo, George C. Gitao, Paul Joseph Plummer, Benard W. Kulohoma, Charles M. Mulei, Rawlynce Bett

**Affiliations:** 1Department of Veterinary Pathology, Microbiology and Parasitology, College of Agriculture and Veterinary Sciences, University of Nairobi, Nairobi P.O. Box 29053-00625, Kenya; cggitao@gmail.com; 2Department of Veterinary Diagnostic and Production Animal Medicine, College of Veterinary Medicine, Iowa State University, Ames, IA 50011, USA; pplummer@iastate.edu; 3Department of Veterinary Microbiology and Preventative Medicine, College of Veterinary Medicine, Iowa State University, Ames, IA 50011, USA; 4National Institute for Antimicrobial Resistance Research and Education, Ames, IA 50010, USA; 5Centre for Biotechnology and Bioinformatics, University of Nairobi, Nairobi P.O. Box 30197-00100, Kenya; bkulohoma@uonbi.ac.ke; 6Department of Clinical Studies, College of Agriculture and Veterinary Sciences, University of Nairobi, Nairobi P.O. Box 29053-00625, Kenya; cmulei@uonbi.ac.ke; 7Department of Animal Production, College of Agriculture and Veterinary Sciences, University of Nairobi, Nairobi P.O. Box 29053-00625, Kenya; rawlynce@gmail.com

**Keywords:** methicillin-resistant staphylococci, bovine mastitis, antibiotic resistance genes, *S. aureus* coagulase-negative staphylococci

## Abstract

Increasing numbers of potentially zoonotic multidrug-resistant (MDR) staphylococci strains, associated with mastitis in dairy cows, are being reported globally and threaten disease management in both animal and human health. However, the prevalence and antimicrobial resistance profiles of these strains, including methicillin-resistant staphylococci (MRS), in Kenya is not well known. This study investigated the drug resistance profiles and genes carried by 183 staphylococci isolates from 142 dairy cows representing 93 farms recovered from mastitis milk of dairy cows in two selected counties in Kenya. Staphylococci isolates were characterized by phenotypic characteristics, polymerase chain reaction (PCR) amplification, partial sequencing and susceptibility testing for 10 antimicrobial drugs. Detection of seven resistance genes to the various antimicrobial drugs was conducted using PCR. Overall, phenotypic resistance among the staphylococci ranged between 66.1% for ampicillin and 3.5% for fluoroquinolones. Twenty-five percent (25%) of *S. aureus* and 10.8% of the coagulase-negative staphylococci (CoNS) isolates, were methicillin-resistant staphylococci phenotypically (defined as resistance to cefoxitin disk diffusion). The most common genes found in *S. aureus* and CoNS were *blaZ* and *strB* at 44.3% and 26%, and 78% and 50%, respectively. MDR was observed in 29.67% and 16.3% of *S. aureus* and CoNS, respectively. These findings pose a threat to bovine mastitis treatment and management as well as human health.

## 1. Introduction

*Staphylococcus aureus (S. aureus)* and coagulase-negative staphylococci (CoNS) are economically significant and potentially zoonotic agents of dairy cow mastitis worldwide [1,2]. Economic losses due to mastitis have been estimated at over USD 2 billion per year in USA [3]. In humans, these Gram-positive bacteria have been isolated from a variety of infections including mastitis, soft tissue infections, food poisoning, endocarditis and septicemia [4,5]. Diseases due to multi-drug resistant (MDR) staphylococci are often severe, challenging to treat, and threaten animal and human healthcare [6].

Antimicrobial treatment is an important tool for controlling intramammary infections in dairy cows in most countries globally [7]. However, the efficacy of antimicrobial therapy against mastitis pathogens including *Staphylococcus* species is on the decline [8]. This low cure rate of staphylococci pathogens is, in part, related to the rise in multi-drug resistance accelerated by over-use and misuse of antimicrobials in veterinary practice [9,10]. Of great concern, is the high level of resistance to β-lactam antibiotics being report among staphylococci species isolated from bovine milk [11,12].

Multidrug-resistant (MDR) staphylococci, including methicillin-resistant staphylococci (MRS), are an emerging global public health problem [13]. Mastitis milk has been documented as an important reservoir of these MDR strains [14,15]. Indeed, the newly emerging livestock-associated methicillin-resistant *Staphylococcus aureus* clonal complex 398 (LA-MRSA CC398) isolated from humans has been linked to cow mastitis milk origin [2,16]. Further, methicillin-resistant coagulase-negative staphylococci (MRCoNS), specifically *Staphylococcus epidermidis,* are increasingly being reported in mastitic milk of cows [17,18]. Increased risk of clonal transmission of MRS between dairy cows and the persons in contact with those animals has also been described [6,10]. This growing trend presents a serious threat to mastitis management and poses a potentially significant public health risk to humans consuming or handling raw milk [10,11].

While poorly quantified, the burden of antimicrobial resistance (AMR) attributable to livestock is comparatively higher in low-income countries, including Kenya and wider sub-Saharan Africa [19,20]. One of the key drivers of this, is the high burden of subclinical mastitis (>50%) and the largely unregulated use of veterinary antimicrobials [20,21]. In addition, Kenyan farmers often self-diagnose and treat cows with mastitis without laboratory confirmation to guide therapy, and there is a general lack of stringent measures on drug withdrawal periods [21,22]. As a consequence, these practices have significantly contributed to the emergence and spread of antibiotic-resistant bacterial strains, resulting in treatment failures [23]. In view of this, there is an urgent need for an evidence-informed policy on antibiotic use and AMR in dairy farming in Kenya [24].

Studies on antimicrobial resistance (AMR) patterns and genes of *S. aureus* are well documented [11,25]. However, to date, only a handful of studies on staphylococcal mastitis pathogens and their associated resistance genes have been conducted in Kenya [26]. The few available studies have mainly focused on phenotypic antimicrobial susceptibility profiles of *S. aureus* with limited attention to genotypic characterization of resistance [27]. Moreover, very few studies on phenotypic and genotypic AMR profiles of CoNS have been conducted in Kenya [28]. Identification of AMR genes in staphylococci species is critical in order to minimize the risk of transmission to persons and animal reservoirs [29]. Therefore, the aim of this study was to determine the antimicrobial susceptibility patterns and the presence of key AMR genes in staphylococci isolated from mastitic dairy cattle in two selected counties in Kenya. This information is key to improving antimicrobial stewardship and to mitigating the emergence and spread of AMR.

## 2. Results

### 2.1. Confirmation of Staphylococci Species with PCR

All 91 (100%) *S. aureus* isolates yielded an amplicon for the *nuc* gene. A subset (eight) of *nuc* genes was sequenced and revealed 97–100% homogeneity to *S. aureus*.

### 2.2. Antimicrobial Resistance Patterns of S. aureus and CoNS

The highest phenotypic resistance was reported in ampicillin (66.1%; *n* = 121) followed by tetracycline (23%; *n* = 42). However, lower resistance rates to fluoroquinolones (4%; *n* = 8) and gentamicin (5.4%; *n* = 10) were noted among the isolates. *S. aureus* showed a significantly higher resistance to cefoxitin and ampicillin compared to CoNS (*p* < 0.009, *p* < 0.014), respectively. However, there was no statistically significant differences in the resistance frequencies for the other antibiotics between *S. aureus* and CoNS.

Among the *S. aureus* isolates, 75.8% (*n* = 69) of the isolates were resistant to at least one the antimicrobial agents tested. As shown in Table 1 ampicillin was the most frequent resistant phenotype (71.4%) followed by erythromycin at 25.2%, while a lower resistance rate was reported in fluoroquinolones (3–4%). Further, 25% of the isolates were phenotypically methicillin-resistant *S. aureus* (MRSA) based on the cefoxitin disc diffusion test. In addition, all MRSA isolates showed 100% phenotypic resistance to ampicillin followed by 52% for erythromycin, 48% for tetracycline, streptomycin 39% and 30% for trimethoprim–sulfamethoxazole. In this study, significantly higher resistance to various antimicrobials was observed in MRSA compared to MSSA (*p* < 0.05), except for ciprofloxacin and trimethoprim–sulfamethoxazole (Table 2).

Among the CoNS, 68.5% (*n* = 63) of the isolates were resistant to at least one antimicrobial agent tested. Ampicillin resistance was the most frequent phenotype at 57.6% followed by resistance to tetracycline (22.8%), while lower resistance to fluoroquinolones (3%) was noted among the isolates. In this study, cefoxitin resistance was reported in 10.8% of the isolates and were termed as methicillin-resistant coagulase-negative staphylococci (MRCoNS) (Table 1).

### 2.3. Prevalence of Multidrug Resistance in Staphylococci Species

Multidrug resistance (MDR), defined as isolates showing resistance to three or more classes of antibiotics, was detected in 22.9% (*n* = 42) of the staphylococci isolates. Significantly, a higher proportion of MDR isolates were reported among *S. aureus* at 29.67% (*n* = 27) compared to CoNS at 16.3% (*n* = 15; *p* = 0.032). Among the *S. aureus,* MRSA and MSSA showed MDR at 56.52% (*n* = 13) and 20.5% (*n* = 14), respectively. For CoNS, 10 isolates classified as MRCoNS, where seven were also MDR. Further, 3.2% (*n* = 6) staphylococci isolates showed resistance to more than seven antimicrobial agents tested, out of which five of them were MRSA and two were MRCoNS.

### 2.4. Detection of Resistant Genes from the Staphylococci Species

Overall, 57.2% (95/166) of isolates harbored at least one resistance gene of the seven different genes detected in different combination. As shown in Appendix A, one isolate carried four genes, eight isolates carried three genes and 20 isolates carried two genes. In addition, 66 isolates carried one gene, of which 97% (64/66) of these isolates carried *blaZ* genes. Notably, 22 of the isolates carrying more than ≥2 genes were MDR. All the isolates carrying ≥2 antimicrobial resistance genes showed high resistance to ampicillin (100%) and tetracycline (75%).

Overall, more resistance genes were reported in CoNS isolates at a prevalence of 70.2% (*n* = 59) compared to *S. aureus*—42.3% (*n* = 36; *p* < 0.001). The most prevalent resistance gene was the β-lactamase gene *blaZ*—59.2% (*n* = 90). The prevalence of the *blaZ* gene was higher in CoNS (65.4%; *n* = 55) compared to *S. aureus* (41.1%; *n* = 35; *p* < 0.002). Strikingly, we noted that 20.5% (*n* = 14) of the MSSA strain carried the *blaZ* gene. Further, one of the MRCoNS isolates carried three resistance genes; namely *blaZ, msrA, strB* conferring resistance β-lactams, aminoglycosides and macrolides (Table 3 and Appendix A).

Tetracycline resistance to *tetM* and *tetK* genes was detected at 16.2% and 6.9%, respectively. We noted, however, that all *tetK* detected in this study were from CoNS, with none from *S. aureus*. The streptomycin-resistant *strB gene* was present in 35.7% of staphylococci isolates. A Higher occurrence of *strB* genes was reported in CoNS at 50% compared to *S. aureus* at 13%. Among the erythromycin-resistant isolates, *msrA* (24.3%) and *ermB* (21.6%) were the most prevalent among the isolates. The frequencies of *msrA* and *ermB* in *S. aureus* and CoNS were similar (Table 3).

## 3. Discussion

Our study investigated the antibiotic susceptibility patterns and antimicrobial resistance genes in staphylococci recovered from mastitic cow milk in Kenya in order to improve mastitis therapy and the control emergence and spread of AMR.

*S. aureus* and CoNS are recognized as the leading causative agents of mastitis in dairy cows in Kenya [21,30]. However, further molecular characterization of the CoNS to species level is required in Kenya in order to confirm the circulating species and their specific antibiogram.

Overall, our study showed a high proportion of resistance (71.5%) to at least one antibiotic tested, with no differences between *S. aureus* (75%) and CoNS (68%). This was slightly lower than reported in other studies in Malaysia (96.15%) and South Africa (90%) in *S. aureus* and CoNS, respectively [31,32] In contrast, a slightly lower resistance proportion in *S. aureus* (50%) and CoNS (50%) were reported in Uganda [33]. High resistance levels observed in this study could be linked to indiscriminate use of antimicrobials for treatment of mastitis in dairy cows by farmers and veterinarians [34]. As with many developing countries, in Kenya most of these antimicrobials are cheap and readily available as over-the-counter drugs and can be bought without any veterinary prescription [33]. However, further studies investigating antibiotic use and practices in dairy farms in Kenya are necessary to provide conclusive evidence on the extent which such practices contribute to the spread of antimicrobial resistance.

This study reported much higher resistance rates in penicillins (ampicillin) than those reported previously in dairy cows in Kenya by Gitau et al. [21] and Shitandi et al. [27] of about 30% [21,27]. However, our findings were comparable to a recent report by Mureithi et al. [28] in Kenya, who found a prevalence rate of 64% in ampicillin [28]. These results indicate an increase in resistance in staphylococci over the years. High resistance levels to penicillin and other β- lactams among mastitis-causing staphylococci has been described [35,36,37]. In contrast, a lower resistance to penicillin in staphylococci (0–20%) has also been reported in some European countries [38,39] and Canada [7]. Higher resistance rates are likely due to the wide-spread use of penicillin in the treatment of mastitis in dairy cows as observed in the study farms and as evidenced by other previous studies in Kenya [27,40] Further, changes overtime, spatial sampling, differences in antibiotic use and practices might explain discrepancies in resistance levels between regions [33,41]. 

We observed that *S. aureus* had a significantly higher proportion of ampicillin resistance when compared to CoNS species. *S. aureus* has been described as a common cause of bovine mastitis in the study region [21]. This higher resistance could be due to the fact that penicillin is still the first-line drug of choice for treatment of mastitis in Kenya [27,34]. Routine culture and identification, coupled with antibiotic sensitivity testing, should be adopted before treatment with antimicrobials to avoid selection pressure of antimicrobial resistant *S. aureus* [37,39]. 

*S. aureus* had a significantly higher proportion of methicillin resistance compared to CoNS in this study. A study in Korea found a slightly higher prevalence of MRSA compared to MRCoNS [42]. However, in contrast, Schnitt and Tenhagen [43] in their review highlighted several studies that have reported higher MRCoNS compared to MRSA in mastitic milk samples [43]. Noteworthy, in all these studies, MRS was defined based on the presence of the *mecA* gene unlike in our study and could explain the discrepancy. However, although the reason for the higher prevalence of MRSA in our findings is unclear; lower virulence observed in MRCoNS might be a contributing factor [43].

Low resistance levels to quinolones and chloramphenicol were reported in this study among the staphylococci. Authors in Ethiopia [44], South Africa [32], Canada [7] and Uganda [33] reported similar findings. These critically important human-medicine antibiotics are restricted for use in treatment of animal diseases in many countries, including Kenya [32]. However, even the low resistance rates reported are of public health significance and control measures should be implemented to curb further spread [7].

In the present study, 25% of *S. aureus* and 10.8% of CoNS isolates were phenotypically resistant to cefoxitin, and were consequently classified as MRSA and MRCoNS, respectively. Cefoxitin disk tests have shown in several studies to be a reliable marker for methicillin-resistant *S. aureus* and CoNS not identified to species levels [45,46]. It is worth noting that all MRSA and MRCoNS showed resistance to ampicillin. Our findings on MRSA were in close agreement with reports by Liu et al. [37] in China. A relatively higher prevalence of cefoxitin-resistant MRSA has been reported in Malaysia (38.6%) and Ethiopia (58.1%) [31,45]. Similarly, a higher prevalence of phenotypic MRCoNS has been reported in South Korea (21.2%) [47], Tunisia (29.41%) [48] and Switzerland (47%) [36]. Detection of methicillin-resistant staphylococci (MRS) in mastitic milk is a public health concern and should be further investigated as most of these organisms have shown to be potentially zoonotic in addition to multidrug-resistant, reducing the role of therapy in the control of staphylococcal mastitis [29]. The culling of infected cows to avoid further transmission would be of highest importance [49].

Presence of the *mecA* gene is considered the gold standard of defining MRSA, in addition to the new resistance genes *mecC* and *mecB* which are homologue to *mecA* [14,46,50,51]. However, in this study, screening for *mec* genes in the phenotypically methicillin-resistant strains were not carried out which is a limitation in this study. Further investigation targeting *mec* genes and other mechanisms should be carried out in future studies in order to broaden our understanding on the genetic basis of antimicrobial resistance of these isolates [52]. Molecular typing to assess the clonality of the isolates is also recommended.

Strikingly, the trimethoprim–sulfamethoxazole resistance rate in MRSA reported in this study was quite high (30%). Higher resistance rates in *Escherichia coli* (*E. coli*) to trimethoprim–sulfamethoxazole have been reported in livestock (54.2%), especially in poultry and pigs in Kenya [53]. Although data supporting the use of trimethoprim–sulfamethoxazole directly in cows are limited, sulfonamides have been reported as the second commonly used antibiotic in food animals after tetracycline [34,40]. Moreover, Muloi et al. [54] in their study on antibiotic practices and knowledge among antibiotics retailers in Nairobi, reported that sulfonamides (63%) were amongst the most purchased class of drug by dairy farmers from Agrovet in Nairobi, Kenya, which neighbors the two study counties [54]. Further, according to Mitema et al. [40] sulfonamides are extensively used in the poultry industry and treatment of calf scours and pneumonia in Kenya [40]. The role of horizontal transfer of AMR genetic determinant between different bacterial species and among humans and animal species has been described [20]. This high resistance rate reported in this study is very concerning since trimethoprim–sulfamethoxazole is heavily used for prophylaxis in HIV-infected patients especially in a highly infectious disease setting, including Kenya [55].

This study reported 20.5% (*n* = 14) of methicillin-susceptible *S. aureus* (MSSA) with an effective *blaZ* variant. A significantly higher prevalence of *blaZ* (91%) in MSSA has been reported in a hospital setting in Kuwait [56]. Qu et al. [8] also found that 76 of *S. aureus* isolates carried the *blaZ* gene but lacked the *mecA* gene [8]. Presence of *blaZ* genes in *S. aureus* has been shown to play a significant role in promoting the acquisition and stabilization of the *mecA* gene [57]. However, according to Vali et al. [56] and Milheiriço et al. [57] the presence of the *blaZ* gene in MRSA and MSSA may be responsible for encoding for resistance to penicillin only [56,57]. Penicillins are extensively used in food animals and are still the first-line drug of choice for the treatment of mastitis in Kenya [27]. Further, African MSSA isolated from hospital settings in urban areas has shown to have a significantly higher resistance to penicillin, ranging between 73.7 and 100% compared to other MSSAs as described by [58] in their review. Diversities between *blaZ* allotypes due to non-clonal evolutions in MRSA and MSSA isolates have also been observed in different geographical regions [57]. Therefore, evaluation of the *blaZ* allele between MRSA and MSSA isolates in the study should be investigated in future.

Similar to other studies, we observed a significantly higher multidrug resistance (MDR) in MRSA strains compared to MSSA [29,37]. MRSA have shown to have the potential to develop resistance to nearly all the antimicrobial agents [2]. This evolving trend, and the rapid emergence of antimicrobial resistance in *S. aureus,* threatens disease management in both animal and human health [10].

A higher proportion of MDR isolates were reported among *S. aureus* compared to CoNS. In contrast, Dorneles et al. [59] and Cheng et al. [60] reported higher MDR rates in CoNS compared to *S. aureus* in a similar study in Brazil and China, respectively. Significantly higher MDR rates in *S. aureus* have been described [29,37]. The widespread resistance of cattle-derived *S. aureus* presents a serious challenge to bovine mastitis therapy and a potential public health risk to humans in Kenya.

Knowledge of distribution of antimicrobial resistance genes among pathogenic udder microbes is key to understanding the evolution of multidrug-resistant agents in dairy cattle. Higher levels of resistance genes were reported in CoNS (70.2%) compared to *S. aureus* (42.3%) in this study. These findings support the hypothesis that CoNS are considered a main reservoir of genetic elements transferrable to other species of bacteria, including *S. aureus* [61].

In agreement with other studies, genes encoding for the beta-lactamases *blaZ* gene were identified in staphylococci at 97% [8,62]. This likely contributed to the high resistance level to penicillin (ampicillin 66.1%) recorded in this study. High *blaZ* genes might indicate an increased use, and possibly misuse, of β lactams in the study farms [62].

We observed a low to moderate prevalence of antimicrobial resistance genes (ARGs) to tetracycline (*tetM and tetK*), macrolides (*msrA, emrB, ermC*) and streptomycins (*strB*) compared to the phenotypic resistance. Low prevalence of ARGs has been reported by Pekana et al. [63] who found low expression of genotypic resistance in *S. aureus* in South Africa. Other studies by Gao et al. [64] and Yang et al. [62]—both in China—also reported low genotypic resistance compared to phenotypic resistance in staphylococci isolates. Resistance mediated by other independent mechanisms, such as point mutations, biofilm formation or antibiotic tolerance, could explain these findings [36,65]. Moreover, resistance genes not included in this study may account for phenotypic resistance observed [63]. Further, use of human disc diffusion interpretative criteria may have contributed to the misalignment between phenotypic and genotypic resistance observed in the isolates. Whole genome sequencing is needed to expand our knowledge on staphylococci and their genetic basis of antimicrobial resistance [66].

## 4. Materials and Methods

### 4.1. Study Areas and Design

Study areas and design have been previously described [30]. Briefly, this study was a cross-sectional survey undertaken between November 2018 and June 2019 in Embu and Kajiado—counties in Kenya. The counties were selected purposefully based on the high populations of cows, the human population’s demand for milk and dairy products, and diversity of cattle breeds and farming practices. Lactating dairy cows of local and exotic origin in different stages of lactation and parity were randomly selected and sampled.

### 4.2. Herd and Sampling

The isolates utilized in this study were obtained from a previous study on prevalence of mastitis in dairy cows in Kenya [30]. Out of the 595 available staphylococci isolates, a total of 183 colonies, including 91 *Staphylococcus aureus (S. aureus)* and 92 coagulase-negative staphylococci (CoNS) from 142 dairy cows representing 93 smallholder farms, were randomly selected and analyzed. Distribution of the isolates among the counties were as follows: For *S. aureus, n* = 58 isolates from 31 cows from Embu and *n* = 43 were isolated from 29 cows from Kajiado County. For CoNS, *n* = 53 isolates from 48 cows in Embu and *n* = 39 were isolated from 34 cows in Kajiado. Detection of clinical and subclinical mastitis was conducted as previously described by [67].

### 4.3. Milk Sample Collection

Milk samples were collected following standardized procedures described by the National Mastitis Council [68]. The samples were transported at 4 °C to the University of Nairobi, Department of Veterinary Pathology, Microbiology and Parasitology bacteriology laboratory for microbiological examination.

### 4.4. Laboratory Analysis

#### 4.4.1. Isolation and Phenotypic Characterization of the Isolates

Initial bacterial culture and identification was performed according to standard methods described in [68,69]. Briefly, a 0.01 mL aliquot of each milk was aseptically streaked onto the surface of 5% sheep blood agar (Oxoid, Basingstoke, England) and incubated aerobically at 37 °C for 24–48 h. Where growth occurred, colony morphology, Gram stain reaction, catalase, coagulase tube testing and mannitol salt agar (Oxoid, Basingstoke, England, UK) were used to presumptively identify *S. aureus* and CoNS. Single colonies from respective isolates were sub-cultured on nutrient agar slants, and the slants were stored at 4 °C for further use.

#### 4.4.2. Bacterial Genomic DNA Extraction

The boiling method was used to extract bacterial genomic DNA [70]. Briefly, a loopful of a bacterial colony grown overnight on tryptone soy agar (TSA) was added to 1.5 mL Eppendorf tubes containing 100 μL of nuclease-free water. The tubes were boiled in water bath at 100 °C for 25 min. After centrifugation at 30,000× *g* for 5 min in a microcentrifuge (Eppendorf Hamburg, Germany), the supernatant was transferred to a new 1.5 microcentrifuge tube. The extracted DNA was stored in a freezer at −20 °C until used for PCR analysis.

#### 4.4.3. Staphylococcus Aureus *nuc* Gene Amplification

PCR amplification of the staphylococcal thermonuclease (*nuc*) gene was used to confirm ninety-one biochemically identified *S. aureus*. The oligonucleotide primers described by [71] in Table 4 were used and the PCR was performed using Taq Polymerase (QIAGEN, GmbH, Hilden, Germany) following the manufacturers’ instruction. For positive and negative controls, *S. aureus* ATCC 29,213 (*nuc* positive strains) [50] and DNase deionized water, respectively, were used. Electrophoresis for each PCR amplicon was performed using ethidium bromide-stained 1.5% agarose gel in the Tris-Borate-EDTA (TBE) buffer and visualized using UV illuminator (GelMax^®^ Imager, Cambridge, UK). A molecular ladder was used to determine the sizes of the amplicon (GelPilot 1 kb Plus Ladder (100), QIAGEN, GmbH, Hilden Germany) with an expected amplicon size of 276 bp.

#### 4.4.4. Antibiotic Susceptibility Testing

Phenotypic antibiotic susceptibility characterization was carried out using the Kirby–Bauer disc diffusion method following the Clinical & Laboratory Standards Institute (CLSI) guidelines [50]. Briefly, fresh cultures of bacterial isolates were suspended in sterile physiological saline to attain turbidity equivalent to 0.5 McFarland. Sterile cotton wool swabs were then used to inoculate the standardized bacterial suspension onto Mueller–Hinton (Oxoid) agar plates. The antibiotics discs belonging to seven classes of antibiotics at the following concentration were analyzed: aminoglycosides (gentamicin 10 µg, streptomycin 10 µg), fluoroquinolone (ciprofloxacin 5 µg and norfloxacin 10 μg), tetracycline (tetracycline 30 μg), folate pathway inhibitors (sulfonamide + trimethoprim 25 μg), macrolides (erythromycin 15 µg), beta lactams (ampicillin 25 µg and cefoxitin 30 μg) and phenicols (chloramphenicol 10 μg). The choice of antibiotics was guided by drugs that are commonly used in dairy veterinary practice in Kenya, some of which are important to human medicine. The inoculated plates were then incubated at 35–37 °C for 17 h.

Since most of the antibiotics used have no approved bovine breakpoints, interpretive criteria described by [46,50] were used. The inhibition zones were measured and interpreted as either susceptible, intermediate, or resistant to the tested antibiotic agent when appropriate break points were available. However, in this study, all strains read as intermediate were considered as resistant. Due to lack of approved interpretive criteria for streptomycin in staphylococci, isolates were interpreted using the interpretive criteria of another aminoglycoside (gentamicin 10 µg) in this study [50]. *Staphylococcus aureus* ATCC 25,923 was used as the quality control strain. Isolates resistant to cefoxitin (*S. aureus*) and (CoNS) were presumptively identified as MRSA and MRCoNS, respectively [46,50]. The isolates were classified as multidrug-resistant (MDR) if they were found resistant to three or more different antimicrobial classes used [75].

#### 4.4.5. Detection of Antimicrobial Resistance Genes

All staphylococci isolates (*S. aureus* and CoNS) showing phenotypic resistance to beta lactams, erythromycin, tetracyclines and streptomycin were analyzed by PCR for genes that confer resistance to penicillin (*blaZ*), erythromycin (*ermB*, *ermC, msrA*), streptomycin (*strB*) and tetracycline (*tetK*, *tetM*). The details on primers and annealing temperatures used in the study are provided in Table 4. PCR products (10 μL) were electrophoresed using ethidium bromide-stained 1.5% agarose gel in the Tris-Borate-EDTA (TBE) buffer and visualized using the UV illuminator (GelMax^®^ Imager, Cambridge UK). Molecular ladder was used to determine the sizes of the amplicon (GelPilot 1 kb Plus Ladder (100), QIAGEN, GmbH, Hilden Germany). In all PCR reactions, RNAse-free water was used as negative control.

### 4.5. nuc Gene and Antibiotic Resistance Gene Sequencing and Analysis

A subset of the PCR amplicon of the *nuc* and the antibiotic resistant genes were sequenced to confirm identities of the detected organisms and genes using Sanger DNA sequencing approaches [76]. Quality control, assembly and editing of nucleic sequence trace files were performed using SnapGene version 5.2.4 [77] and customized UNIX shell scripts. Sequence identities were confirmed using the Basic Local Alignment Search Tool (BLAST) [78].

### 4.6. Statistical Data Analysis

Antibiotic resistance data were analyzed using STATA version 15. Descriptive statistics were used to calculate the proportion and frequencies of all variables. The chi-square test (χ^2^ test) or Fisher’s exact test were used when applicable to compare categorical variables. Statistical significance level was set at 0.05 (*p* < 0.05).

## 5. Conclusions

This study revealed a high ampicillin resistance rate among bovine mastitis staphylococci. In addition, detection of various antimicrobial resistance genes in these strains signifies a public health concern and a serious challenge to bovine mastitis therapy. Therefore, there is a need to control the emergence and spread of AMR in dairy farms. The presence of phenotypic methicillin-resistant staphylococci (MRS) in this study provides baseline for further monitoring in Kenyan dairy farms. Further screening of the *mec* genes (*A, B, C*) and other intrinsic mechanisms encoding for resistance to MRS should be considered in future studies. Molecular characterization of CoNS isolates to species level will be necessary to confirm circulating species in Kenya. We used PCR to determine the genotypic resistance, this technique targets fewer AMR genes, restricting the results to screened elements. Therefore, there is a need for further characterization of the isolates using whole genome sequencing and spa typing in order to assess the clonal diversities of the isolates.

## Figures and Tables

**Table 1 antibiotics-10-00772-t001:** Antibiotic susceptibility patterns of 183 staphylococci isolated from mastitic cow milk in two counties, Kenya.

Antibiotic Class	Disk Concentration (µg)	Disc Diffusion Interpretive Criteria (mm)		*S. aureus*	CoNS
		S ^1^	R ^2^	R*n* (%)	R*n* (%)
β-lactams					
Cefoxitin	30	≥22	≤21	23 (25)	10 (10.8)
Ampicillin	25	≥29	≤28	65 (71.4)	53 (57.6)
Aminoglycosides					
Gentamicin	10	≥15	≤12	6 (6)	4 (4.3)
Streptomycin	10	≥15	≤12	23 (21)	18 (20)
Fluoroquinolones					
Ciprofloxacin	5	≥21	≤15	3 (3.2)	3 (3)
Norfloxacin	10	≥17	≤12	4 (4.3)	3 (3)
Tetracycline					
Tetracycline	30	≥19	≤14	23 (21)	21 (22.8)
Folate pathway inhibitors					
Trimethoprim–sulfamethoxazole	23.75/1.25	≥1	≥10	17 (17.5)	16 (17.3)
Macrolides					
Erythromycin	15	≥18	≤13	23 (25.2)	14 (15.2)
Phenicols					
Chloramphenicol	10	≥18	≤12	8 (8.7)	7 (7.6)

^1^ sensitive; ^2^ resistant. Disc diffusion interpretive criteria for cefoxitin in CoNS was performed based on EUCAST 2021 (S ≥ 25, R < 25) and the rest according to CLSI M100 2016.

**Table 2 antibiotics-10-00772-t002:** Antimicrobial resistance pattern of MRSA and MSSA isolated from mastitic cows in two counties, Kenya (*n* = 91).

Antimicrobial Agents	MRSA ^1^ (*n* = 23)	MSSA ^2^ (*n* = 68)	*p*-Value ^3^
	R*n* ^4^ (%)	R*n* (%)	
Cefoxitin	23 (100)	0 (0)	0.001
Ampicillin	23 (100)	45 (66.2)	0.001
Gentamicin	5 (26)	1 (1.4)	0.04
Norfloxacin	3 (13)	1 (1.4)	0.04
Streptomycin	9 (39)	12 (17.6)	0.03
Ciprofloxacin	1 (4)	2 (2.9)	0.58
Trimethoprim–Sulfamethoxazole	7 (30)	10 (14.7)	0.06
Tetracycline	11 (48)	10 (14.7)	0.006
Erythromycin	12 (52)	12 (17.6)	0.001
Chloramphenicol	6 (26)	2 (2.9)	0.001

^1^ Methicillin-resistant *Staphylococcus aureus*; ^2^ Methicillin-sensitive *Staphylococcus aureus*; ^3^ *p*-value refers to differences between MRSA and MSSA isolates resistant to the respective antimicrobial drug; ^4^ Resistance.

**Table 3 antibiotics-10-00772-t003:** Antimicrobial resistance genes to various antibiotics among the staphylococci isolates from bovine mastitis in two counties in Kenya.

Species	β-Lactams		Tetracycline			Streptomycin		Erythromycin			
	^1^ No. R	*blaZ* R*n* (%)	No. R	*tetM* R*n* (%)	*tetK* R*n* (%)	No. R	*strB* R*n* (%)	No. R	*msrA* R*n* (%)	*ermB* R*n* (%)	*ermC* R*n* (%)
*S. aureus*	79	35 (41.1)	23	4 (17.3)	-	23	6 (26)	23	5 (21.7)	4 (17.3)	-
CoNS	73	55 (65.4)	21	3 (14.2)	3 (14.2)	18	9 (50)	14	4 (28.5)	4 (28.5)	1 (4.3)
Total	152	90 (59.2)	43	7 (16.2)	3 (6.9)	42	15 (35.7)	37	9 (24.3)	8 (21.6)	1 (2.7)

^1^ Number of phenotypic-resistant isolates in each category.

**Table 4 antibiotics-10-00772-t004:** Details of primers and annealing temperatures used to detect antibiotic resistance genes in the study.

Target Gene	Primer Sequence (5′-3′)	Annealing Temperature (°C)	Amplicon Size (bp)	Reference
*nuc*	F-GCGATTGATGGTGATACGGTT	50	276	[71]
	R-CAAGCCTTGACGAACTAAAGC			
*blaZ*	F-ACTTCAACA CCTGCTGCTTTC	54	173	[72]
	R-TGACCACTTTTATCAGCAACC			
*strB*	F-CGGTCGTGAGAACAATCTGA	60	313	[73]
	R-ATGATGCAGGATCGCCATGTA			
*ermB*	F-ACGACGAAACTGGCTAA	55	409	[64]
	R-TGGTATGGCGGGTAA			
*msrA*	F-AAGGCTTGTCCGCAATACAC	60	320	[73]
	R-CCATTACCCCCAATAAGTGC			
*tetM*	F-GTCCGTCTGAACTTTGCGGA	59	662	[26]
	R-GCGGCACTTCGATGTGAATG			
*tetK*	F-TTAGGTGAAGGGTTAGGTCC	59	718	[26]
	R-GCAAACTCATTCCAGAAGCA			
*ermC*	F-AATCGGCTCAGGAAAAGG	55	562	[74]
	R-ATCGTCAATTCCTGCATG			

F—forward; R—reverse.

## Data Availability

All datasets are available from the corresponding author on reasonable request.

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
