# Peer review of "Antimicrobial Resistance Profiles and Genes of Staphylococci Isolated from Mastitic Cow’s Milk in Kenya"

_antibiotics, 2021, doi:10.3390/antibiotics10070772_

Round 1

Reviewer 1 Report

The manuscript entitled „Antimicrobial Resistance Profiles and Genes of Staphylococci Isolated from mastitis cow milk in Kenya” aimed to highlight the problems of antimicrobial resistance and therapy of staphylococcal mastitis in cow in Kenya. The topic of the paper is current, and the authors have indeed presented important results worth considering for publication.

However, there are several limitations in the current state of the paper that need to be addressed.

The main problem is associated with significant discrepancies between the results of mecA gene detection and screening with cefoxitin. Resistance to cefoxitin was detected in 23 S. aureus isolates, and mecA only in one. Isolates with reduced susceptibility to oxacillin / cefoxitin and other beta-lactams are commonly referred to as BORSA (borderline oxacillin-resistant S. aureus) and are extremely rare in widespread distribution. The revealed fact of the prevalence of BORSA isolates against the background of almost complete absence of MRSA requires detailed discussion.

The section of antibiotic susceptibility testing needs revision. The authors should carefully study current recommendations of EUCAST and CLSI for susceptibility testing of staphylococci and perform necessary tests. Below are the main, but not all notes:

  • Neither EUCAST, nor CLSI recommend oxacillin disk-diffusion for S. aureus susceptibility testing (screening).
  • Neither EUCAST, nor CLSI recommend disk-diffusion for vancomycin susceptibility testing. Recommendations given in reference 36 are outdated. It is better to exclude data from the article.
  • Different approaches are recommended for screening of different CoNS species. Accordingly, the CoNS should be identified to the species level.

The discussion requires serious revision, considering the comments made.

Reviewer 2 Report

The authors performed an epidemiological study on S. aureus and CoNS of bovine origin in Kenya. In total 183 isolates were studied, originating from 142 animals. The study premise is interesting and relevant in light of the one health concept.

You may want to consider these comments to improve your manuscript.

Specific comments:

  1. Line 91: Was all 91 sequenced or just a subset? This was not clear.
  2. Line 94. I suggest removing the first sentence as it is not really relevant… phenotypical resistance to which antibiotics? Consider removing this sentence altogether.
  3. Lines 102-105: how can you explain the discrepancy of the oxacillin testing with the cefoxitin test results. There are some oxacillin-resistant MSSA with a more effective blaZ variant. This is usually considered rare. This discrepancy should be discussed in the discussion.
  4. Only a small percentage of your MRSA actually harbor a mec gene. Molecular typing, e.g. with spa-typing would be interesting to assess the clonality of the isolates. Although this has been described before, this is quite rare as mec gene is considered the hallmark of MRSA. Not so long ago, mecB has been reported as a mechanism of methicillin resistance (MRSA) in S. aureus. If molecular detection for mecB was not performed, you should at least mention this in the discussion as a limitation.
  5. Line 121: what is your definition of MDR. Usually resistance to 3 or more classes of antibiotics. This should be stated in the manuscript.
  6. Line 345: did you mean ATCC25923 instead of ATCC29923?
  7. The high-percentage of phenotypic oxacillin resistance in MSSA is surprising, since you would expect that MSSA would be oxacillin susceptible. Did you perform the oxacillin phenotypic testing on salt-supplemented Mueller Hinton Agar? The salt concentration can influence the phenotypic testing. This discrepancy was not mentioned in the discussion, this should be added.
  8. The trimethoprim-sulfamethoxazole resistance in MRSA is quite high (30%). In Africa, a lot of clinical (human) isolates are resistant to this substance combination. This should be included in the discussion. Besides human medicine (TMP-SMZ is used heavily in HIV patients), this substance is also used in pig farming. Is TMP-SMZ widely used in cattle farming?
  9. Table 1: genes should be in italics
  10. Table 2: a high rate of VRSA is unusual. This might need explanation or confirmation by vanA gene detection.
  11. Table 3: I presume that sulfamethoxazole is referring to trimethoprim-sulfamethoxazole in this Table? If yes, please correct.

Round 2

Reviewer 1 Report

Dear Editor, The authors have significantly improved the article. In my opinion, the article can be published in its present form